# Immunosuppressive Tumor Microenvironment and Immunotherapy of Epstein–Barr Virus-Associated Malignancies

**DOI:** 10.3390/v14051017

**Published:** 2022-05-10

**Authors:** Xueyi Zheng, Yuhua Huang, Kai Li, Rongzhen Luo, Muyan Cai, Jingping Yun

**Affiliations:** 1Department of Pathology, Collaborative Innovation Center for Cancer Medicine, State Key Laboratory of Oncology in South China, Sun Yat-sen University Cancer Center, Guangzhou 510060, China; zhengxy1@sysucc.org.cn (X.Z.); huangyh@sysucc.org.cn (Y.H.); luorzh@sysucc.org.cn (R.L.); 2Department of Liver Surgery, Collaborative Innovation Center for Cancer Medicine, State Key Laboratory of Oncology in South China, Sun Yat-sen University Cancer Center, Guangzhou 510060, China; likai@sysucc.org.cn

**Keywords:** Epstein–Barr virus, tumor microenvironment, EBV-associated malignancies, immune evasion

## Abstract

The Epstein–Barr virus (EBV) can cause different types of cancer in human beings when the virus infects different cell types with various latent patterns. EBV shapes a distinct and immunosuppressive tumor microenvironment (TME) to its benefit by influencing and interacting with different components in the TME. Different EBV-associated malignancies adopt similar but slightly specific immunosuppressive mechanisms by encoding different EBV products to escape both innate and adaptive immune responses. Strategies reversing the immunosuppressive TME of EBV-associated malignancies have been under evaluation in clinical practice. As the interactions among EBV, tumor cells, and TME are intricate, in this review, we mainly discuss the epidemiology of EBV, the life cycle of EBV, the cellular and molecular composition of TME, and a landscape of different EBV-associated malignancies and immunotherapy by targeting the TME.

## 1. Introduction

Epstein–Barr virus (EBV), the first discovered human tumor virus, was isolated in Burkitt lymphoma samples by Michael Anthony Epstein and Yvonne Barr in 1964 [1,2]. This offers important insights into the natural infectious history of EBV and the mechanisms of how this virus drives the oncogenic process. EBV, also known as human herpesvirus 4, a member of the gammaherpesvirus subfamily [3], is characterized by the most common virus infection in humans, with a prevalence of more than 95% in the human population worldwide [4]. This virus normally persists as a lifelong infection without symptoms in humans, thus striking a subtle balance between the virus, the host cell, and the host immune system [5].

EBV, with a diameter of about 150–170 nm, consists of about 170 kb double-stranded DNA, which encodes more than 85 genes [6]. The viral genome controls EBV infectious mechanisms and contributes to the initiation and symptoms of EBV-associated human diseases. However, the particular function of 30% to 40% of the EBV genes remains unknown. There are two major EBV types, type 1 and type 2, which are mainly different in the genes encoding Epstein–Barr nuclear antigen 2 (EBNA2) and EBNA3 genes [7,8]. Type 1 is more prevalent worldwide and potentially transmits to B cells with a higher transforming efficiency while type 2 is mainly located in Africa [9].

Like other herpesviruses, EBV can build lifelong latency after its primary infection. EBV in teenagers usually causes symptomatic infectious mononucleosis but in adults the primary EBV infection is asymptomatic [10]. However, people with immunosuppressed conditions can also suffer severe acute diseases after EBV infection [11] and life-threatening EBV-associated malignancies originating from the lymphoid and epithelial cells [12,13]. It is reported that EBV causes more than 200,000 cases of cancer each year and 1.8% of cancer-related deaths [14,15]. Males encounter a higher prevalence of EBV-attributable malignancies compared to females [16]. Geographically, EBV-associated malignancy is present all over the world but is more prevalent in Asia and Africa than in the Western world [15]. It is well known that EBV can transform tumor precursor cells into EBV-associated malignancies [9,14].

In addition, EBV can also shape the supportive and immunosuppressed tumor microenvironment (TME) for its oncogenic process [17,18]. Available evidence indicates that the microenvironment plays an important role in the development of EBV-associated malignancies and the complex cross-talk among EBV and the microenvironment is not only relevant for establishing persistent EBV infection but also associated with the development of EBV-driven tumors and keeping the subtle EBV–host balance [19,20].

## 2. Viral Life Cycle and Gene Expression

The EBV life cycle in human beings often starts with its access to naive B cells in the oropharyngeal mucosa following transmission through the saliva. In the process of primary infection, EBV-infected naive B cells lead to a latency III growth program, which proliferates and expands the infected cells with no epigenetic repression of latent EBV genes [21]. Therefore, these cells express all the latent genes, including EBNAs, latent membrane protein (LMPs), EBV-encoded small RNAs (EBERs), and microRNAs (miRNAs) [22,23]. Being highly immunogenic, these cells can be recognized by the host immunity and rapidly eliminated by EBV-specific T cells [24]. Primary infection with EBV is followed by asymptomatic persistence of the virus in memory B cells where EBV gene expression is probably completely silenced. In this progress, the infected B cells epigenetically repress most genes to avoid the immune attack with a very high level of CpG methylation of its DNA [25]. The mechanism by which EBV gains access to the memory B cell pool is controversial. The most widely held model proposes that EBV-infected memory B cells are generated via a germinal center reaction. In this process, the virus enters the default program in the germinal center [26]. B cells in the default program are characterized by the expression of latency II proteins, including EBNA1, LMP1, and LMP2. These proteins are similar to the same signals as antigen-driven B cell responses in the normal germinal center [27,28]. Then, EBV-infected B cells leave the germinal center and gain entry into the memory B cell pool. These cells shut down all the viral genes (latency 0) that encode viral proteins and mimic the same signals inducing the proliferation of normal memory B cells. To allow the viral genome to replicate along with the cell, the EBV-infected memory B cells need to express EBNA1 (latency I), which can bind with the origin of plasmid replication (oriP), the viral plasmid origin of DNA replication [29,30]. Memory B cells can differentiate terminally into plasma cells, which is regarded to be associated with the switch to the EBV lytic program [31]. The lytic program can be further divided into three different lytic phases, including immediate early, early, and late phases. This process is characterized by the expression of up to 80 lytic genes [32]. During EBV lytic cycle, the viral chromatin opens up, which initiates the replication of the viral gene and synthesizes the full cascade of viral proteins [33,34]. The newly synthesized viral DNA, which lacks histones and methylated CpG dinucleotides, is packaged into viral capsid structures [35]. Then, the newly produced virions are released from the infected cell and, in the process, destroy the cells. Then, the virions may infect new target cells such as B cells and epithelial cells, where the virus can further replicate and then be transmitted to new hosts. Like the above replication program, the EBV life cycle oscillates between lytic phases and latent phases in infected cells [36].

Although it is not well understood how the virus establishes its latency and switches to a lytic program in the target cells, the progress has been investigated intensively and is thought to be well regulated by epigenetic regulation. As a rule, the EBV’s genomic DNA in infected cells is presented as extrachromosomal plasmids. A powerful epigenetic repression of lytic genes ensures the latent pattern of the EBV-infected cells, whose expression of different latent viral genes depends on different cell types. Epigenetic regulation also controls the progress from latency III to latency I, which can be regulated by CpG methylation of viral DNA, polycomb group proteins, and high-density packaging of nucleosomes [25].

Disruption of this well-regulated B cell infection can result in EBV-associated B cell tumors in all latency types except latency 0 [22]. The role of EBV in the oncogenic process of epithelial tumors, such as undifferentiated nasopharyngeal carcinoma (NPC) and EBV-associated gastric carcinoma (EBVaGC), remains unknown but is considered as a result of the aberrance of viral latency in epithelial cells with the existence of premalignant gene changes [37,38]. The progress and distinguished gene expression profiles of B cells infected by EBV are summarized in Figure 1.

## 3. TME of EBV-Associated Malignancy

Some EBV-associated malignancies develop after chronic latent EBV infection. It has been well known that tumor cells can be recognized and eliminated by the innate and adaptive immune system. However, the development of EBV-associated malignancies is tightly associated with the TME shaped by tumor cells to suppress the hosts’ immune system and evade immune surveillance. Indeed, tumor cells can turn immune cells to a tolerogenic and exhausted state by different components in the TME, including cell–cell interactions, soluble molecules, physical molecules, and extracellular vesicles (EVs) (Figure 2).

### 3.1. Cellular Components

Immune cells are the major components in the TME of EBV-associated malignancies. These cells try to clear and kill the EBV-infected malignant cells on the one hand and establish an immunosuppressive environment to provide local support for the tumor on the other hand.

#### 3.1.1. T Cells

Antivirus responses depend heavily on potent virus-specific T cells. The EBV resides in the host cell most of the time after the establishment of virus infection. T cells recognize and eliminate EBV-infected cells based on virus-derived peptides presented on the surface of HLA molecules. The numbers of different T cell subsets vary greatly between individuals.

CD8+ T cells are the most powerful immune cells targeting the EBV-infected cells and executing the antiviral immunity. As HLA class I molecules (HLA I) are expressed in all nucleated human cells, antiviral CD8+ T cells will detect and target the EBV-infected cells. Compared to EBV-negative tumors, the number of CD8+ T cells is increased in EBV-positive tumors. Higher numbers of CD8+ T cells are associated with a higher frequency of effector T cells expressing the cytotoxic molecules. To evade recognition by EBV-specific CD8+ T cells, EBV encodes products that independently interfere with antigen presentation on HLA I by shutting off the expression of HLA I molecules, downregulating the HLA I on the cellular surface, and blocking the process of antigen presentation in the cell [5]. A recent study showed high numbers of exhausted CD8+ T cells were present in both the TME and the peripheral blood of EBV-positive tumors, resulting in the reduced cytotoxic activity of CD8+ T cells [39]. The exhausted CD8+ T cells are induced by EBV in several ways, including increasing checkpoint molecules, higher levels of immunosuppressive cytokines, and downregulation of death receptors, e.g., FasL and TRAIL.

CD4+ T cells are an indispensable component of the adaptive immunity against EBV, especially when EBV-infected cells express a high level of HLA II. These CD4+ T helper (Th) cells, CD4+ Th1 cells, in particular, can specifically target and eliminate EBV-infected cells either directly through their cytotoxic capacity or indirectly through activating CD8+ cytotoxic T cells and B cells. In response, EBV adopts varying immune escape strategies interfering with CD4+ T cell immunity at different levels. Gp42 is initially described as an entry receptor for EBV to bind to HLA class II molecules present on B cells [40]. Furthermore, it also serves as an immunosuppressive molecule interacting with HLA class II/peptide complexes to block the function of T cell receptors and prevent the activation of CD4+ T cells [41]. The early host shutoff protein BGLF5 encoded by EBV decreases cell-surface HLA II by degradation of HLA II mRNAs [42]. EBV also encodes a viral IL-10 homolog BCRF1, acting as an anti-inflammatory cytokine that inhibits and modulates the function of CD4+ T cells [43].

In the peripheral blood and tumor tissues of patients with EBV-positive tumors, an increasing number of natural CD4+CD25+ regulatory T cells (Tregs) have been found [44,45]. The viral latent protein EBNA1 plays a direct role in recruiting natural Tregs by upregulating CCL20 chemokine [46], and LMP1 can also serve as a peptide to promote infiltration of Treg cells [47]. A recent study showed that EBV-infected cells increased the infiltration of regulatory Type 1 cells (Tr1), which upregulate the gene expression of Tr1-related markers (ITGA2, ITGB2, LAG3) and associated immunosuppressive cytokines (IL-10) [48]. This upregulation is associated with an overexpression of several chemokine markers, such as CCL17, CCL19, and CCL20, which are known to attract T-helper type 2 (Th2) and regulatory T cells, thus contributing to more powerful immunosuppression.

#### 3.1.2. Tumor-Associated Macrophages (TAMs)

The TME of EBV-positive tumors is also characterized by a more pronounced infiltration by TAMs as compared to EBV-negative cases. Interacting with the various signals in the TME, TAMs can show polarized activation into two functional states, including the M1 macrophages, which present a proinflammatory phenotype to promote Th1 responses and kill tumor cells, and the M2 macrophages, which can have an immunosuppressive role to promote tumor growth by remodeling and promoting Th2 responses. The macrophages are more likely to polarize into M1 macrophages to keep pace with a predominant Th1 microenvironment in EBV-positive tumors [49]. However, compared to EBV-negative tumors, EBV-positive tumors still have significantly higher numbers of M2 macrophages [50], which are mainly distributed in the stroma and involved in tumor progression [51]. This type of macrophage is attracted and recruited to the TME by chemokines such as CCL2 and CCL5, which can be induced by the EBV latent protein LMP1 in an NF-kappa B (NF-κB)-dependent manner [50].

#### 3.1.3. Dendritic Cells (DCs)

DCs are the most potent antigen-presenting cells in the initiation of antiviral immune responses [52,53]. In EBV-associated malignancies, high levels of DCs are found to infiltrate into tumor regions [54,55]. However, the activation and maturation of DCs into different functional subsets depends largely on the cytokine milieu in the TME. EBERs within exosomes could promote DC activation and subsequent T cell activation, resulting in the systemic production of proinflammatory cytokines. EBV exosomes containing deoxyuridine triphosphate nucleotidohydrolase (dUTPase) can regulate the innate immune function of DCs through toll-like receptor (TLR)-2 leading to NF-κB activation and the production of proinflammatory cytokines, thus modulating the cellular microenvironment [56]. To evade the host immune responses, the EBV-infected cells suppress the function of DCs in several ways. The maturation of DCs is suppressed by galectin-1, a carbohydrate-binding protein within exosomes from EBV-infected cells [57]. Peripheral DCs infiltrate the tumor and are induced to differentiate into LAMP3+ DCs [58], which express PD-L1/PD-L2 interacting with PD-1 on CD8+ T cells and thus promote CD8+ T cell exhaustion [59].

#### 3.1.4. Myeloid-Derived Suppressor Cells (MDSCs)

MDSCs originate from immature myeloid cells which are incapable of differentiating into mature myeloid cells, such as DCs, macrophages, and granulocytes [60]. MDSCs are expanded and induced by LMP1 in EBV-positive tumors of latency type II and III [61], exhibiting a strong immunosuppressive function. The LMP1 regulates the production of IL-1β, IL-6, and GM-CSF to enhance tumor-associated MDSC expansion [61]. MDSCs can support the naive CD4+ T cells to differentiate into Tregs by secreting retinoic acid and TGF-β and promote the transdifferentiation of Th17 cells into Foxp3+ Tregs [62]. Additionally, MDSCs can also trigger immunosuppressive functions of Tregs through mediating the release of IL-10 and IFN-γ [63]. Tumor-infiltrating MDSCs also present high levels of CCR5 ligands to recruit high numbers of Tregs into the TME [64], establishing an additional interaction between MDSCs and Tregs. Therefore, the MDSCs play an important role in the establishment of an immunosuppressive TME in EBV-positive tumors.

#### 3.1.5. NK Cells

NK cells are the one arm of the innate immune response against EBV and have an important antiviral function during primary infection [65,66]. Although NK cells are highly infiltrated in EBV-positive malignancies, their functions can be impaired to various degrees when EBV-infected cells enter the latency phase [67]. LMP1 induces the expression of several antiapoptotic proteins such as Survivin, A20, Bcl2 [68], and a prosurvival receptor, 4-1BB [69,70], in EBV-infected cells to resist infected-cell death and apoptosis induced by NK cells via death receptors, which is regarded as the main mechanism of how NK cells kill the EBV-infected cells [71]. LMP2A and LMP2B can impair the antiviral response against EBV-infected cells by downregulating IFN signaling [72], and subsequently weaken cytokine-mediated antiviral effects. The EBV-encoded miRNAs, such as miRNA-BARTs, can dull NK cell immune recognition, and impair NK cell cytotoxic response by diminishing IL-12 release [73,74]. In such ways, the EBV weakens the antitumor function of NK cells and contributes to immune evasion in EBV-positive malignancies [75].

#### 3.1.6. B Cells

Upon activation, B cells can differentiate into plasma cells and secrete antibodies to tag pathogens or infected cells for immune destruction. However, few studies have been focused on the function of B cells within the TME of EBV-associated malignancies. It is reported that B cells infiltrate into many EBV-positive tumors, but with a lower frequency than T cells do. Moreover, the number of infiltrated B cells varies greatly and they mostly reside in the stroma [76]. LMP1 can block the differentiation of B cells into antibody-secreting cells [77] through the upregulation of indoleamine 2,3-dioxygenase 1 (IDO1). IDO1 can also inhibit B cell function, thus further suppressing immune responses [77]. It is reported that EBV-infected cells can express miRNA-21 and then induce the expression of IL-10 in naive B cells, which is capable of suppressing CD8+ T cell activities [78].

#### 3.1.7. Cancer-Associated Fibroblasts (CAFs)

CAFs, a major component of the cancer stroma derived from normal fibroblasts, can be induced and activated by LMP1 in EBV-infected cells through the NF-κB or ERK-MAPK pathway [79,80]. These CAFs recruited to the TME can promote cancer progression [81], facilitate metastasis by secreting proteases to degrade the extracellular matrix, provide support, fuel tumors’ insatiable growth by releasing paracrine cytokine [82], and recruit other stromal cell types with immunomodulatory properties [83]. The interplay between EBV-infected cells and CAFs forms the immunosuppressive microenvironment.

#### 3.1.8. Endothelial Cells

Endothelial cells are involved in cancer pathogenesis and progression in various processes, including inflammation, fibrinolysis, and angiogenesis [84]. In EBV-positive tumors, LMP1 can induce the expression of fibroblast growth factor 2 (FGF2) and vascular endothelial growth factor (VEGF) to activate the endothelial cells to support tumor neoangiogenesis [85]. Furthermore, EBERs can be detected in endothelial cells in EBV-infected tissues, implicating that EBERs released from EBV-infected cells can act on endothelial cells and promote angiogenesis by stimulating VCAM-1 expression [86]. Innate immune modulation induced by EBV lytic infection can also promote endothelial cell inflammation and vascular injury [87].

### 3.2. Molecular Components

The local secretion of proinflammatory and immunosuppressive cytokines and chemokines is induced by EBV-infected cells. In turn, these aberrant soluble components also affect the function of immune cells and thus shape a TME where EBV-infected cells can proliferate, escape from apoptosis, and survive host antitumor defense [88].

#### 3.2.1. Soluble Molecules

As for proinflammatory function, IFN-γ is upregulated in EBV-positive tumors. The cytokine IFN-γ, released by immune cells, such as T cells and DCs which can be recruited by EBV-infected cells, plays a central role in the resistance of the host to EBV infection via direct antiviral effects as well as modulation of the immune response. IFN-γ can directly enhance the macrophage’s differentiation into M1-like macrophages to establish an inflammatory TME. The BLZF1 encoded by EBV-infected cells seems to inhibit the function of the IFN-γ signaling pathway by downregulating the IFN-γ receptor and the downstream effector of IFN-γ. IP-10, a 10 kDa IFN-γ-inducible protein, is upregulated in many EBV-positive tumors. IP-10 is a member of the CXC chemokine family which is involved in chemotaxis, induction of apoptosis, regulation of cell growth, and mediation of angiostatic effects [89]. It is reported that the expression of IP-10 can be also mediated by LMP1, independent of IFN-γ, not only through transcriptional but also post-transcriptional mechanisms [90]. IL-1β is found to be upregulated in EBV-positive tumors [91]. The upregulation of IL-1β is induced by EBV genomic DNA and EBERs [92]. Tumor-derived IL-1β inhibits tumor growth and enhances survival through host responses. Mechanistically, IL-1β-mediated antitumor effects depend on infiltrated immunostimulatory neutrophils, which are significantly associated with better survival in patients with EBV-positive tumors [92]. TNF-α can be upregulated by LMP1 through TRAF2,5 and the NF-κB pathway in EBV-infected cells [93]. TNF-α may induce apoptosis and cell injuries via binding to TNF-α receptor-1 (TNFR1) [94]. However, these LMP1-expressing EBV-infected cells are resistant to TNF-α-induced apoptosis by downregulating TNFR1 and suppressing the downstream activities of apoptotic caspases 3, 8, and 9 [93].

Besides proinflammatory soluble molecules, immunosuppressive cytokines are found more frequently in EBV-positive tumors than in EBV-negative tumors. IL-10 is a potent immunosuppressive cytokine that induces Tregs and inhibits Th1 cells and cytotoxic T lymphocytes (CTLs) [95,96]. The expression of IL-10 is higher in EBV-positive tumors compared to EBV-negative tumors [96,97]. Higher levels of IL-10 are associated with lower numbers of CTLs in EBV-associated malignancies [98]. In EBV-infected cells, LMP2A can induce the expression of IL-10 [99,100]. IL-10 can also be produced by DCs, Tr1 cells, TAMs, and MDSCs to play the immunosuppressive role in EBV-associated malignancies. It is reported that higher levels of IL-4, IL-6, and IL-13 are present in EBV-positive tumors acting as a growth factor [101] and induce expression of the EBV-encoded protein LMP1 [102]. In turn, the induced LMP1 may directly enhance the production of immunosuppressive cytokines such as IL-6 and IL-8 via an NF-κB pathway [103,104]. The interplay between EBV-infected cells and immunosuppressive cytokines contributes to establishing a beneficial environment for tumor growth. Moreover, the expression of galectin-1 can be enhanced by both LMP1 and LMP2 proteins to induce tolerogenic DCs and cause the selective apoptosis of CD4+ T cells and CTLs [105].

#### 3.2.2. Checkpoint Molecules

Immunomodulatory molecules such as PD-1 and its ligands PD-L1 and PD-L2 have been carefully investigated in EBV-associated malignancies. They play an indispensable role in assisting tumor cells to escape the host immune system [106]. Anti-PD-1 antibodies have been tested and applied in clinical practice and showed a remarkable response in EBV-associated malignancies [107,108]. Indeed, LMP1 can upregulate the expression of PD-L1 in EBV-infected cells through activation of TLR signaling and STAT3 or the engagement of AP-1-associated enhancers [109,110]. The induced expression of PD-L1 can be further enhanced by stimulation with IFN-γ [111]. Moreover, EBV-specific T cells can be also induced to express higher levels of PD-1 to further suppress immune responses against EBV-associated malignancies [112]. T-lymphocyte antigen-4 (CTLA-4) is another important immune checkpoint that is upregulated in T cells of EBV-infected tissues. High expression of CTLA-4 is significantly associated with disease progression and worse overall survival in patients with EBV-positive tumors [113]. Furthermore, other immunomodulatory molecules, such as TIM-3, LAG-3, and VISTA, are upregulated in EBV-specific T cells, which can further impair the function of T cell targeting the EBV [114,115].

### 3.3. EVs

Oncogenic viruses shape a beneficial protumoral milieu to promote the development and progression of malignancies [116,117]. EVs are a major component to facilitate intercellular communications in the TME. EVs can be divided into three major groups, including exosomes (30–150 nm in diameter, inward budding from the endosomal membrane), microvesicles (50–1000 nm in diameter, outward budding directly from plasma membranes), and apoptotic bodies (800–5000 nm in diameter, released directly from the plasma membrane of apoptotic cells), according to their production mode [118]. However, due to the limitations of methods to purify the EVs, it is almost impossible to isolate the pure subtypes and study their distinct functions. Since exosomes are the most frequently studied EVs in the EBV research field, we will mainly discuss the exosomes in the TME of EBV-associated malignancies.

EVs can transmit bioactive molecules enclosed in a lipid bilayer [119] to the TME to contribute to physiological and pathological processes by influencing recipient cells [117]. EBV is capable of modifying EVs’ content and influencing their release, thus enhancing the interplay between EBV-infected cells and immune cells in the TME. EBV can package various viral products, such as viral proteins and RNAs into the EVs [120] to aid in the evasion of the host immune system and promote tumor progression within the TME [120,121] (Figure 3).

LMP1 has been identified in exosomes. This latent EBV protein, when taken up by recipient cells, enables the induction of cellular responses by inhibiting apoptosis and promoting cell growth. LMP1 also supports tumor progression by mediating the function of immune cells in the TME. LMP1 can act on B cells and subsequently suppress humoral immune responses by preventing B cells’ differentiation into antibody-secreting cells [77]. LMP1 can upregulate the expression of IDO1 and then inhibit B cell function in the surrounding cells, thus suppressing immune effects [77]. Interestingly, LMP1 is found to regulate PD-L1 [122] and increase the packaging of PD-L1 into exosomes [123] in EBV-infected cells, which is involved in immune suppression. Moreover, exosomes containing LMP1 can induce the expression of angiogenic factors FGF2 and VEGF [124,125], which can enhance the proliferation of blood vessel endothelial cells and promote angiogenesis in tumor development. Therefore, the LMP1 released in EVs can modulate the immune responses and contribute to an immunosuppressive TME.

LMP2 is found to be secreted from EBV-infected cells in exosomes [126]. Though the biological function of LMP2 extracellular vesicles targeting recipient cells is not well understood, exosomal LMP2A may mimic the B cell receptor signal and inhibit B cell activity. LMP2A can promote the EBV life cycle by providing growth signals in the recipient cells while LMP2B has a distinct function in recipient cells and activates B cells [127].

EBERs are noncoding RNAs that can be found in all EBV-infected cells, thus regarded as the gold standard to identify the EBV infection in clinical practice. EBERs can be packaged with the EBER binding protein La into exosomes [128]. Evidence showed that EBERs can inhibit apoptosis, increase cell proliferation, and induce tumor formation [129,130]. EBERs are found to promote the release of IFN-I and inflammatory cytokines through TLR-3-mediated signaling [131]. Therefore, EBERs circulating within exosomes could trigger DC activation and subsequent T cell activation, leading to the systemic production of proinflammatory cytokines.

The miRNAs can be packaged into EVs and secreted into the TME for interaction with other cells [132]. There are 44 mature EBV miRNAs, including four of BamHI fragment H rightward open reading frame 1 (BHRF1), and forty BamHI-A rightward transcripts (BARTs) [133]. EBV miRNAs can suppress the secretion of proinflammatory cytokines, which can further suppress CD4+ T cell differentiation and reduce the activation of cytotoxic EBV-specific CD4+ effector T cells [134]. EBV-infected cells also reduce the functions of CD8+ T cells by releasing miRNAs that directly influence the process of antigen presentation and reduce MHC class 1 molecules present on the cellular surface. The miRNAs can also reduce EBNA1 levels, which is a target of CD8+ T cells, and therefore reduce the recognition and cytotoxic effects of CD8+ T cells [134].

EBV exosomes also contain dUTPase, a protein encoded by the EBV lytic gene BLLF3, which can influence the TME by modulating both innate and adaptive immune responses [56]. In particular, exosomes containing EBV dUTPase were shown to enhance the production of inflammatory cytokines and induce NF-κB activation by interacting with DCs through TLR-2 [56].

Galectin-1 is found to be packaged in EBV exosomes. Galectin-1 can affect local immune responses by promoting the apoptosis of Th1, Th17, and CTLs [135] and inducing the tolerant state of DCs [136].

Galectin-9, a β-galactoside-binding protein, can be derived from EBV exosomes in NPC patients. It can be induced to release by EBV products and interact with LMP1 [137]. These galectin-9-containing exosomes are shown to have an immunoregulatory function, by inducing apoptosis in EBV-specific CD4+ T cells through galectin-9/Tim-3 interactions [138] and expanding the number of Tregs and enhancing their suppressive activities [139].

Although the immunosuppressive role of EVs creates obstacles in controlling the tumor, EVs released by EBV-infected cells are now becoming more universally accepted as a superior source of biomarkers for diagnosing cancer due to their greater stability and higher diagnostic performances. In the case of EBV-associated malignancies, EVs containing EBV miRNAs have been investigated and proposed as early diagnostic markers in NPC and gastric cancer [140,141]. In addition, EVs are also considered as a promising immunotherapy target against EBV-associated malignancies for their immunosuppressive role by carrying different EBV products. A recent study showed that exosomes derived from Vδ2-T cells could promote the expansion of EBV-specific CD4+ and CD8+ T cells and control EBV-associated tumor cells through FasL and TRAIL pathways [142]. However, further studies need to be done to investigate the potential role of exosomes as specific therapeutic targets against EBV-associated malignancies. For example, preventing EBV-encoded products from loading into exosomes or inhibiting the release of such exosomes might serve as novel strategies [143].

## 4. Landscape of Different EBV-Associated Malignancies

All EBV-associated malignancies can be regarded as rare accidents of the normal EBV life cycle. Although these diseases have similar oncogenic routes, different EBV-associated malignancies are distinguished from each other in epidemiology, histologic appearance, EBV positive rates, latency pattern, and TME. The TME of EBV-associated malignancies is characterized by the coexistence and subtle balance of proinflammatory and immunosuppressive factors. These malignancies adopt similar but slightly specific immunosuppressive mechanisms to evade the host immune responses. Here, we describe the epidemiology, TME, immune evasion strategies, and immunotherapy of different EBV-associated malignancies (Table 1).

### 4.1. EBV-Associated Lymphomas

#### 4.1.1. Post-transplant Lymphoproliferative Disorder (PTLD)

PTLD, a wide range of heterogeneous lymphoproliferative diseases from benign lymphoproliferations to malignant lymphomas, originates from B cells and is involved in 2%-10% of solid organ and hematopoietic stem cell transplants. These diseases are associated with EBV and are mainly characterized by a latency III pattern [144].

In the setting of acquired immune deficiency, EBV can fully exploit all its capabilities to promote the growth of the infected B cells and resist apoptosis. LMP1 is the main driver for B cell growth and survival by increasing the expression and secretion of cytokines and chemokines, such as IFN-γ, IL-6, and IL-10 [145,146]. LMP1 can also modulate the expression of some genes resisting apoptosis, including c-FLIP and BCL-2 [147,148], and upregulate the antiapoptotic Bcl-2 family member Mcl-1 through miRNA-155 expression [148]. Higher levels of IL-10 can also be induced by LMP-2A in B cells [149]. EBNA2, the main driver of type III latency, also enhances the growth and survival of B cells by reducing the expression of the proapoptotic BIK protein [150]. Besides latent proteins, uncontrolled EBV lytic replication may also contribute to the development of EBV-associated lymphoproliferations. A viral IL-10 homolog encoded by EBV lytic genes can also be expressed by EBV-infected cells to interfere with the host cytokine milieu. BZLF-1, the main EBV lytic transactivator, can upregulate the production of cytokines, such as IL-6, IL-10, and IL-13, to promote the growth of B cells [151]. PTLD contains sizeable T cell populations, mainly consisting of the memory/helper type while cytotoxic T cells are strikingly low in all samples [152].

Adoptive T cell therapy has been successfully applied for treating patients with EBV-associated PTLD by reconstituting host cellular immunity against the disease. The first clinical trial of adoptive T cell therapy showed that infusion of donor-derived EBV-specific T cells (EBVSTs) into patients with allogeneic hematopoietic stem cell transplantation (HSCT) induced complete regression in all five patients, but graft-versus-host disease developed [153]. Later, in vitro expanded donor-derived or autologous EBVSTs in phase I clinical trials were proven effective in the prevention and treatment of PTLD in HSCT or solid organ transplantation patients with minimal alloreactivity [154,155].

#### 4.1.2. Hodgkin Lymphoma

The EBV-positive classical Hodgkin lymphoma, involved in approximately 50% of classical Hodgkin lymphoma, is a particular type of lymphoma with a specific TME, which is made up of a small proportion of Hodgkin and Reed–Sternberg (HRS) cells (<1%) and a large number of infiltrating immune cells (>90%) [156,157]. Therefore, the HRS cells greatly depend on interactions with immune cells in the TME by recruiting and communicating with various cellular components through autocrine and paracrine signals [18].

Classical Hodgkin lymphoma is distinguished by the expression of TNFR family members and their ligands. In addition, this lymphoma produces unbalanced Th2 cytokines and chemokines [158], thus leading to the reactive infiltration of eosinophils, Th2 cells, and fibroblasts [159]. These reactive cells also promote the growth of HRS cells. Such Th2 cytokines and chemokines can also attract Treg cells to suppress the EBV-specific immune responses [44]. The EBV-positive Hodgkin lymphoma is a tumor of type II latency pattern, expressing a series of latent genes, including EBNA1, LMP1, LMP2, EBER, and BART RNAs [160]. ENBA1 can further upregulate the expression of the chemokine CCL20 in the HRS cells and thus increase the migration and recruitment of Tregs to the TME. LMP1 epitopes are also found to promote infiltration of Treg cells into EBV-positive Hodgkin lymphoma [46,47]. In contrast, activated CD8+ T cells are scarcely found in areas that are abundant in tumor cells, suggesting that they may be ineffective when targeting the EBV-positive HRS cells. The HRS cells can induce the exhaustion of CD8+ T cells through the secretion of IL-10, TGF-β, and gelatin-1. In addition, PD-L1 is highly expressed in EBV-positive HRS cells, further inhibiting the function of CD8+ T cells by inducing their exhaustion [161].

Since PD-L1 plays an important role in the immunosuppressive TME of classical Hodgkin lymphoma, PD-1 inhibitors, such as nivolumab and pembrolizumab, have been tested in clinical trials and have shown to be effective in patients with classical Hodgkin lymphoma [107,108]. Clinical trials with in vitro expanded EBVSTs targeting type II latency antigens significantly increased response rates as well as overall survival in HL patients [162].

#### 4.1.3. Burkitt Lymphoma (BL)

BL was the first human tumor found to be associated with a particular virus, EBV. Based on geographic distribution, BL is classified into three subtypes: endemic BL, sporadic BL, and immunodeficiency-associated BL. About 95% of endemic BL are EBV positive. In contrast, only less than 15% of sporadic BL and 40% of immunodeficiency-associated BL are associated with EBV.

Different from Hodgkin lymphoma containing at least 90% immune cells, BL seems to lack such a supportive TME. EBV-associated BL cells usually show the restricted type I latency pattern. EBERs can induce the secretion of the anti-inflammatory cytokine IL-10 by directly interacting with RIG-I and IRF3 activation [163]. To promote B cell growth, EBER2 can also induce the expression of cytokine IL-6 and EBNA1 can upregulate IL-2 receptors and enhance the antiapoptotic effects of IL-2 [164]. Higher levels of TAMs, M2-TAMs in particular, are the most prominent in the TME in BL [165]. They might play an important role in tumor progression through the secretion of chemokines, cytokines, and the expression of immune checkpoint-associated proteins such as PD-L1 [165].

With its rapid growth, BL is a highly chemotherapy-sensitive disease and was one of the early cancers in which cures were achieved with chemotherapy alone [166]. Therefore, fewer studies have been focused on immunotherapy against BL. It has been reported in a study that PD-1 antibody showed no objective response in patients with BL [167].

#### 4.1.4. Diffuse Large B Cell Lymphoma (DLBCL)

DLBCL accounts for 30% of all lymphomas and is the most common B cell lymphoma. Approximately 10% of cases of DLBCL are associated with EBV [168]. Compared to the EBV-negative DLBCL, EBV-positive DLBCL has a more unfavorable prognosis. DLBCL, with the tumor cell content ranging from 60% to 80% [169], is characterized by a diffused growth pattern of large CD20+ B cells [170]. Cases arising in elderly patients tend to display latency III infection and cases in younger patients more frequently present a latency II infection pattern.

EBV-associated DLBCL expresses higher levels of PD-L1 compared to EBV-negative tumors, implicating an immune-tolerant mechanism of this tumor [171]. Increased PD-L1 expression in EBV-associated DLBCL cell lines can further induce PD-1 expression on T cells in vitro, subsequently exhausting CD8+T cells [172]. The expression of immunosuppressive cytokine IL-10 is increased in EBV-associated DLBCL [173], which can be induced by LMP1 [172] or Treg cells [17]. Although CD8+ T cells are increased [173], they might not be sufficient to eliminate EBV-infected cells in EBV-associated DLBCL because fewer central and effector memory CD8+ T cells are found among patients with EBV-positive DLBCL [174]. The immunosuppressive TME is also associated with the increased expression of other immune checkpoints, such as LAG3 and TIM3, and a greater protumoral M2 TAM polarization pattern [175]. Therefore, although characterized by inflammatory infiltration, EBV-associated DLBCL may still represent an ineffective immune attempt to control EBV-infected tumor cells [18].

Although the majority of patients with DLBCL can be cured with the standard immunochemotherapy R-CHOP, one-third of them relapse with a dismal outcome in most cases [176]. Chimeric antigen receptor (CAR) T cell therapy has emerged as a treatment option to treat these refractory DLBCLs. From early reports to multicenter studies, the clinical efficacy of CAR T cell therapy has been proven in these refractory patients [177]. As the PD-1/PD-L1 pathway is upregulated in EBV-associated DLBCL, it is expected that PD-1/PD-L1 blockade may improve the potency of CAR T cell therapies [178].

#### 4.1.5. Extranodal NK/T Cell Lymphomas

Extranodal NK/T cell lymphomas are frequently EBV positive, usually affecting immunocompetent patients, although NK cells or T cells are the noncanonical cellular target of EBV. With a type II latency pattern, extranodal NK/T cell lymphomas are characterized by obvious inflammatory infiltration, implicating the supportive role of the inflammatory environment in the development of this tumor. 

Extranodal NK/T cell lymphomas are often localized in the nasal area, where inflammation often occurs and B cells, T cells, and NK cells are thus recruited. The NK cells and T cells might acquire EBV infection when they kill EBV-infected B cells, which might produce the virus in the process. LMP1 expressed by the EBV-infected cells promotes the proliferation of the tumor, which is further potentiated by the cytokine IL-2 secreted by immune cells in the TME [179]. In turn, LMP1 can also increase the sensitivity of the cell to IL-2. Furthermore, activated T cells and macrophages produce IL-10 to enhance the tumor-growing function of IL-2. The sera levels of CD27 are increased in patients with NK/T cell lymphomas, which can interact with the CD70 receptor expressed in the EBV-infected NK/T cells to mediate tumor growth [180]. The EBV-infected T cells can upregulate the expression of TNF-α and promote the proinflammatory process. To keep an immunosuppressive TME, the LMP1 expressed in EBV-infected T cells can confer resistance to TNF-α-induced apoptosis through downregulating TNFR-1 [181]. NK/T cell lymphoma cells express PD-L1, which can be upregulated by LMP1 through the MAPK/NF-κB pathway [182]. Ligation of PD-1 on T cells with PD-L1 on lymphoma cells inhibits the function of T cells, providing a potential mechanism for NK/T cell lymphoma cells to evade immunosurveillance.

In stage I/II NK/T cell lymphoma, combined chemotherapy and radiotherapy is the best approach [183]. In relapsed/refractory cases, blockade of PD-1 has recently shown promising results with high response rates [184]. Further clinical studies remain to be done to prove the efficacy of immunotherapy in NK/T cell lymphoma.

### 4.2. NPC

The association between EBV and the undifferentiated NPC was found accidentally and was the first discovery that EBV infection is not restricted to B cells [185]. This tumor is characterized by a massive lymphocytic infiltration giving the tumor with a lymphoepithelial-like (LEL) appearance and its unique geographical distribution with high incidence throughout South-East Asia. Regardless of geographical distribution, all cases of undifferentiated NPC worldwide are EBV positive, with every malignant cell harboring EBV and presenting the viral gene [186].

It has been investigated for a long time how EBV triggers the pathogenesis of NPC [187,188]. Whole-genome sequencing has been performed to show the complex genomic changes in NPC. The latent pattern of NPC is best described as latency I/II, an intermediate form with the expression of EBNA1, the noncoding EBERs and miRNAs, and LMP2 [188]. Besides diverse gene profiles through genome sequencing analysis [189], NPC also shows another dimension of heterogeneity, diverse immune cells shaping the beneficial and immunosuppressive TME for tumor growth. EBV-positive NPC cells release diverse cytokines and chemokines to recruit multiple immune cells from the peripheral blood [190]. Naive CD8+ T cells infiltrate into the tumor and then develop into the effector CD8+ T cells and further exhausted CD8+ T cells. EBV-positive NPC cells express HLA-G, inhibiting the cytotoxic function of T cells and NK cells [39]. Peripheral DCs infiltrate into the tumor and then differentiate into LAMP3+ DCs [58]. The mature LAMP3+ DCs expressing PD-L1/PD-L2 can interact with PD-1 on CD8+ T cells, restraining the activation of CD8+ T cells and promoting their exhaustion [59]. The antigen presentation process of DCs will be limited when LAMP3+ DCs interact with Treg cells through CTLA4-CD80/CD86. In the process, the DCs secret exosomes containing IDO1, which enables them to recruit Treg cells and induce their proliferation and activation [139]. The intensive cell–cell interactions in the TME foster an immunosuppressive niche for the tumor growth of NPC.

To activate the immune cells to eliminate the NPC cells, clinical trials have been carried out to evaluate the effect of the monoclonal antibody against human PD-1. Recently, a phase III trial showed that the addition of toripalimab to the standard first-line chemotherapy can provide superior progression-free survival in patients with recurrent or metastatic NPC [191]. In addition, most therapeutic EBV vaccines have been focused on patients with NPC, and early clinical trials were done with DC-based EBV vaccines and recombinant viral vector vaccines [192,193]. Although the therapeutic EBV vaccines have shown increasing T cell responses in patients with NPC, future studies remain to be done to overcome the limited antigen epitopes delivered by DC-based EBV vaccines and the decreased antiviral vector immune responses after repeated immunizations [153,194].

### 4.3. EBVaGC

EBV is associated with approximately 10% of gastric cancer worldwide [195]. Compared to their EBV-negative counterparts, EBVaGC is more common in males and more likely to occur earlier in life, but have a relatively favorable prognosis [196]. When EBV infects normal gastric epithelial cells, EBV-infected cells grow clonally, resulting in the development of EBVaGC. The developmental processes of EBVaGC often carry out the sequence of gastritis–infection–cancer [197,198]. EBV infection occurs in gastric epithelial cells in the setting of the mucosa of atrophic gastritis. EBV-infected lymphocytes are recruited to the gastric mucosa and infect the epithelial cells. During this process, EBV establishes latent infection in the epithelial cells and these inflammatory signals play an important role in clonal growth in infected cells. In the setting of an inflammatory environment, the infected epithelial cells induce epithelial–mesenchymal transition and resistance against apoptosis of founder clones [199,200].

Gastric cancer can be divided into LEL-type tumors and conventional-type tumors based on histological appearance. The LEL-like gastric cancer resembles NPC, which is rich in lymphocytic LEL-like infiltration. These LEL-like tumors, accounting for only 1–4% of all gastric cancer, are associated with EBV in more than 90% of cases [201]. The remaining conventional-type tumors are EBV positive in 5–15% of cases. Viral gene expression in tumor cells is consistently either latency type I or an intermediate latency I/II in more than 50% of EBV-positive cases where LMP2 can be detected [202]. The TME plays a critical role in the pathogenesis of EBVaGC. The microenvironment of EBVaGC is characterized by the massive infiltration of CD8+ T cells [203,204]. These tumor-infiltrating lymphocytes (TILs) create an immune-active TME by eliminating EBV-infected malignant cells. However, there are also specific mechanisms that help EBV escape the host immune responses. The EBV-infected malignant cells express PD-L1 and then recruit PD-L1-positive immune cells to escape the host immune system [205]. In parallel with the immunosuppressive function of PD-L1, Tregs and TAMs are also recruited to remodel the TME of EBVaGC [206,207]. IDO1 is upregulated in EBVaGC, serving as a powerful inhibitory immune molecule reducing the immune responses [208]. In addition, EBVaGC cells release exosomes that contain some EBV-specific molecules and incorporate them with immune cells, contributing to the immunosuppressive TME [125].

To invert the immunosuppressive TME of EBVaGC, phase II clinical trials are underway to recruit patients who are stratified based on EBV status to assess the response to PD-1 inhibitors [209]. A phase I trial is assessing the safety of CRISPR-Cas9-mediated PD-1 knockout in EBV-specific CTL cells in stage IV GCs [209].

### 4.4. EBV-Associated Intrahepatic Cholangiocarcinoma (EBVaICC)

Studies of EBVaICC are so rare that the TME of EBVaICC has not been well characterized. A recent study reported that 6.6% of ICC are EBV positive [210]. EBVaICC is positive for EBNA1, but negative for LMP1 and EBNA2, suggesting that EBVaICC is a type I latency pattern. ICC can be divided into the LEL subtype and the conventional subtype. The LEL subtype of ICC has a close relationship with EBV infection, with 45% of cases of EBVaICC being the LEL subtype, while only 0.7% of non-EBVaICC belong to the LEL subtype. In terms of TME, the densities of tumor-infiltrating immune cells were significantly increased in EBValCC compared with non-EBVaICC. The TILs in EBVaICC mainly consist of CD3+ T cells, CD20+ B cells, and CD68+ TAMs. Among the T cell population, CD8+ T cells accounted for the predominant components. As for TAMs, M1 TAMs are more predominant than M2 TAMs in EBValCC [210]. Based on the PD-L1 expression and tumor-infiltrating CD8+ T cell densities, it is recommended to categorize the ICCs into four TME types: Type I (PD-L1+/CD8-High), Type II (PD-L1−/CD8-Low), Type III (PD-L1+/CD8-Low), and Type IV (PD-L1−/CD8-High). The Type I subgroup shows the best survival while the Type III subgroup had the worst survival. EBVaICC is significantly associated with Type I because 90% of EBVaICCs belonged to Type I [210]. Therefore, patients with EBVaICC are more likely to benefit from anti-PD-1/L1 therapy.

### 4.5. EBV-Associated Smooth Muscle Tumor (EBV-SMT)

EBV-SMT is a rare smooth muscle tumor (<1%) that occurs in patients with any immunocompromised conditions, including transplant patients with long-term immunosuppressive treatment, and patients suffering from human immunodeficiency virus infection or congenital immunodeficiency syndromes [211]. Although the exact mechanism of tumorigenesis is unclear, this tumor is associated with EBV infection and its tumor transformation of smooth muscle cells. EBV-SMT tends to be multifocal and arises in any anatomical location [212]. The prognosis is excellent, with patients dying of opportunistic infections but not the direct effects of the tumor [213].

Histologically, EBV-SMT is recognized as a relatively well-differentiated tumor, presenting with prominent intratumoral T lymphocytes, a low level of mitotic activity, and primitive round cell areas [214]. The tumor is reported to be latency pattern III by expressing EBNA1, EBNA2, LMP1, and LMP2A [215] or a variant of latency type III through the expression of EBNA2 or EBNA3 but no LMP1. This finding determines the mechanism of immune regulation in EBV-SMT because EBNA2 can be highly recognized by cytotoxic T cells [216,217]. Immune reconstitution can help T cells to kill tumor cells, thus contributing to tumor shrinkage [217]. Therefore, although the treatment of EBV-SMT mainly focuses on surgical resection and radiation therapy, immune reconstitution is a potentially promising treatment strategy [218].

## 5. Immunotherapy by Targeting the TME

Different types of strategies have been developed to target the TME by reactivating antiviral immune response and subverting the immunosuppressive microenvironment.

### 5.1. Immune Checkpoint Inhibitors

Expression of PD-L1 has been detected in 90% of EBV-positive HL and NPC as well as a wide range of EBV-associated malignancies including extranodal NK/T cell lymphoma, diffuse large B cell lymphoma, and PTLD [219]. Furthermore, PD-1 is also upregulated in EBV-positive tumors. Therefore, these diseases may be beneficial to immune checkpoint inhibitors that target the PD-1/PD-L1 axis [220]. Immune checkpoint inhibitors could impair inhibitory receptors of the immune cells, and thus convert the exhausted state of T cells and restore the cytotoxic EBV-specific T cell activity against malignant cells. PD-1 and PD-L1 inhibitors (e.g., pembrolizumab, nivolumab, atezolizumab, and durvalumab), as well as CTLA-4 inhibitors (e.g., ipilimumab and tremelimumab), are currently under evaluation and some have been proven effective in clinical trials aiming to reverse the immunosuppressive TME [107,108,221].

### 5.2. T Cell Therapy

EBV-associated malignancies are associated with the latent life cycles of EBV, but the pattern of latency depends on the type of tumor. The viral antigens expressed in various EBV-positive tumors can provide immune-based therapies with target antigens. In immunocompromised patients with EBV-positive tumors of latency III pattern, EBVSTs have shown outstanding success in rapidly restoring EBV-specific immunity [222]. In immunocompetent patients with EBV-positive tumors of latency II pattern, these EBVSTs seem to be less effective because intricate immune evasion strategies have been developed in these patients. However, EBV-positive tumors are frequently associated with exhausted CD8+ T cells and thus some clinical studies have shown that EBVSTs can prolong overall survival in patients with more extensive NPC [155,223]. To ensure clinical efficacy, additional therapies, such as checkpoint inhibitors or EBV vaccines, can be combined with EBVSTs to overcome immune evasion strategies of EBV-associated malignancies [224]. Additionally, gene modifications of EBVSTs may also be useful to reverse the function of inhibitory molecules and provide resistance to growth-promoting genes.

### 5.3. Therapeutic EBV Vaccine

The immune system plays an important role in controlling tumors. Immune-based tumor-specific therapy is under evaluation in clinical studies because it is highly effective with limited adverse effects [225]. Therapeutic EBV vaccines are designed to stimulate the existing or trigger novel antiviral immune responses in patients with EBV-associated malignancies [153,194]. The targets of EBV therapeutic vaccines are mainly the latent proteins of EBV, such as EBNA1, LMP1, and LMP2. These proteins are expressed in most EBV-associated malignancies and play a key role in the transformation of normal cells into tumors [225,226,227]. Most therapeutic EBV vaccinations concentrate on patients with NPC [194]. DCs, derived from autologous monocytes of NPC patients and pulsed with LMP2 epitopes, can activate and boost the EBV-specific CD8+ T cell responses [228]. Recombinant viral vector vaccines were later developed to provide a wide range of epitopes to the host immune system to improve the vaccines’ efficacy [229].

## 6. Conclusions

Although it has not been completely understood how EBV initiates infection and develops EBV-associated tumors, it is well recognized that EBV can shape a beneficial and supportive environment by encoding and releasing EBV products, thus intensively interacting with immune cells in the TME. The TME in an already progressing tumor may also facilitate EBV infection in some cases such as EBVaGC, and the EBV-infected cells are capable of forming an immunosuppressive milieu by adopting a variety of immune evasion strategies. Different EBV-associated malignancies have different types of EBV latent patterns, thus having a similar but slightly different interaction with cells in the TME. Different treatment strategies have been under evaluation to reverse the immunosuppressive TME. Immune checkpoint inhibitors have been proven effective in clinical trials in EBV-associated malignancies. However, one single therapeutic strategy might not be enough to control the disease. Therefore, the combination of different therapies targeting the TME might achieve a synergistic effect but needs further clinical evidence.

## Figures and Tables

**Figure 1 viruses-14-01017-f001:**
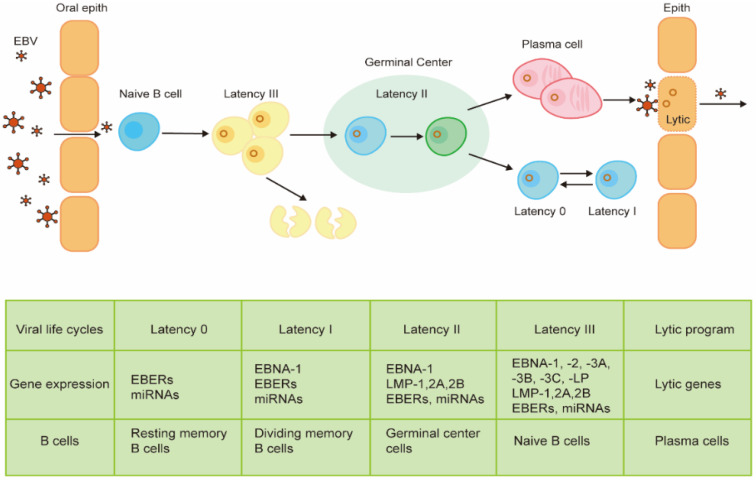
Progress and distinguished gene expression profiles of B cells infected by EBV. The germinal center model is the widely held model to show how the virus enters memory B cells. Orally acquired virus particles may initially infect oropharyngeal B cells, resulting in EBV-transformed B cell growth through a growth program by expressing all the viral proteins (latency III), including EBNA1, EBNA2, EBNA3A-3C, EBNA-LP, LMP1, LPM2A, and LMP2B. Being highly immunogenic, these cells can be recognized and rapidly eliminated by host immunity. Then, the virus-infected B cells go through a physiologic germinal center reaction, by limiting protein expression of latency II (EBNA1, LMP1, LMP2A, and LMP2B). After that, B cells leave the germinal center and enter the memory B cell pool by shutting down all the viral genes (latency 0). When the viral genome replicates along with the cell, EBV-infected memory B cells need to express EBNA1 (latency I). The latently infected memory B cells occasionally reactivate and differentiate into plasma cells, which is associated with the initiation of the lytic program and production virions.

**Figure 2 viruses-14-01017-f002:**
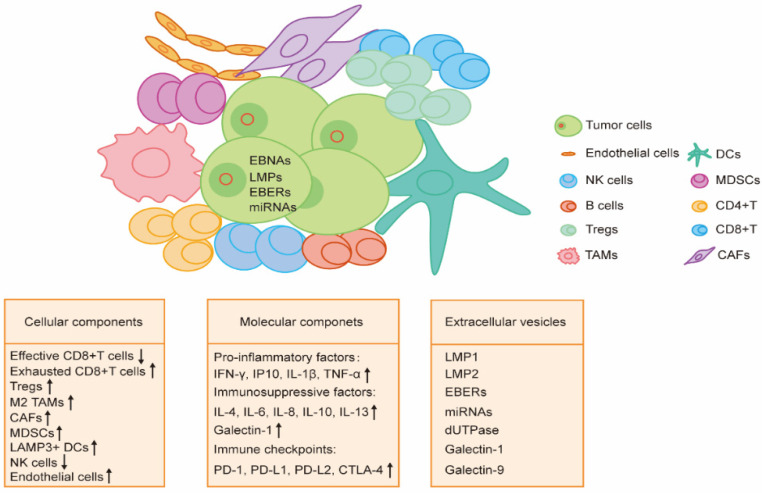
Components of tumor microenvironment in EBV-associated malignancies. EBV-associated malignancies are characterized by the distinct components of the tumor microenvironment. To suppress the host’s immune system and evade immune surveillance, the EBV-infected malignant cells shape and establish an immunosuppressive tumor microenvironment by communicating the cellular components, altering the molecular factors, and releasing extracellular vesicles. TAMs, tumor-associated macrophages; CAFs, cancer-associated fibroblasts; MDSCs, myeloid-derived suppressor cells; DCs, dendritic cells.

**Figure 3 viruses-14-01017-f003:**
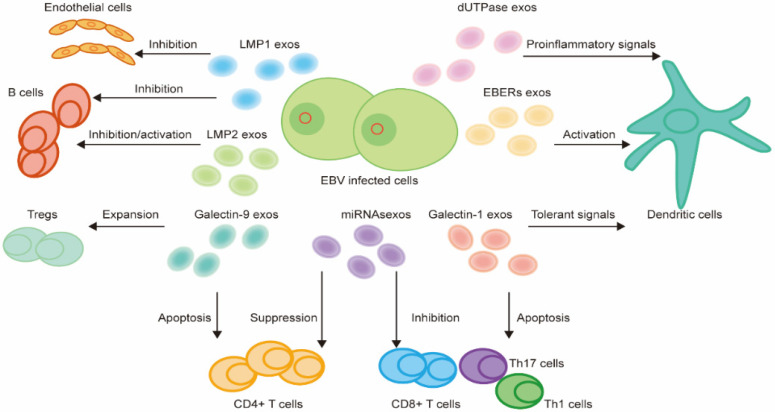
Extracellular vesicles released by EBV-infected cells. The EBV-infected cells release exosomes (exos), containing different components, such as LMP1, LMP2, EBERs, miRNAs, galectin-1, galectin-9, and dUTPase. These extracellular vesicles interact with different immune cells to form a supportive and immunosuppressive tumor microenvironment.

**Table 1 viruses-14-01017-t001:** Features of EBV-associated malignancies.

EBV-Associated Malignancies	EBV Positive Rate	Latency Pattern	Immune Markers		Immunotherapy
			Immune Cells	Immune Molecules	
PTLD	100%	Latency III	Memory/helper T cells Decreased cytotoxic T cells	IFN-γ, IL-6, IL-10, IL-13	Adoptive T cell therapy
Classical HL	50%	Latency II	Tregs Exhausted CD8+ T cells	PD-L1, TNFR, Th2 cytokines and chemokines, IL-10, galectin-1, TGF-β	PD-1 inhibitors T cell therapy
Epidemic BL	100%	Latency I	M2 TAMs	IL-2, IL-6, IL-10	NA
DLBCL	10%	Latency II or III	M2 TAMs Exhausted CD8+ T cells	IL-10, PD-1, PD-L1, PD-L2, LAG3, TIM3	Chimeric antigen receptor T cell therapy
Extranodal NK/T cell lymphomas	100%	Latency II	Activated T cells and macrophages	IL-2, IL-10, CD27, TNF-α, PD-L1	PD-1 inhibitors
Undifferentiated NPC	100%	Latency I/II	Exhausted CD8+ T cells LAMP3+ DCs	PD-L1, PD-L2, CTLA-4, IDO1, HLA-G	PD-1 inhibitors Therapeutic EBV vaccines
GC	10%	Latency I or I/II	CD8+ T cells Tregs TAMs	PD-L1, IDO1	PD-1 inhibitors
ICC	6.6%	Latency I	CD8+ T cells M1 TAMs CD20+ B cells	PD-L1	NA
SMT	<1%	Latency III	T cells	NA	NA

Note: PTLD, Post-transplant lymphoproliferative disease; HL, Hodgkin lymphoma; BL, Burkitt lymphoma; DLBCL, diffuse large B cell lymphoma; GC, gastric carcinoma; NPC, nasopharyngeal carcinoma; ICC, intrahepatic cholangiocarcinoma; SMT, smooth muscle tumor; NA, not applicable.

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
