# Peer review of "Immunosuppressive Tumor Microenvironment and Immunotherapy of Epstein–Barr Virus-Associated Malignancies"

_viruses, 2022, doi:10.3390/v14051017_

Round 1
Reviewer 1 Report
The review ‘Immunosuppressive tumor microenvironment and immunotherapy of Epstein-Barr virus-associated malignancies’ by Zheng et al. is very well structured and looks appealing for the readers who want to broadly understand the relationship between EBV and TME. The review consists of a basic overview of EBV, its genes, and their roles in transformation and tumor progression. The authors have also described different approaches to target TME in EBV-associated malignancies by immunotherapy. Overall, this is a good review and is publishable on the journal. However, some statements and discussions which I think might not be relevant should be revised before the paper is accepted.
- Some of the statements are still very naive and need to be changed. One of the such example is- lines 41 and 58- which somehow seem contradictory to each other. The lines can be either combined or modified as- EBV in teenagers causes symptomatic IM but in adults the primary EBV infection is asymptomatic.
- There is a lot of discussion on EBV latency and lytic phases in this review but the epigenetic regulations controlling the EBV gene expression in different phases are missing. A brief paragraph on EBV epigenetic regulation would be helpful for the readers to understand how the virus infection progresses from Latency III to I, accompanied by differential gene expression.
- In section 6, the authors conclude that EBV can develop tumors and affect TME with induction in proinflammatory cytokines. The authors can consider stating that the TME in an already progressing tumor may also facilitate EBV infection in some cases such as EBVaGC.
Some less major but not minor comments,
- Line 45 and 46 are missing references.
- Line 81; seems vague as the cause of the EBV lytic switch is still debatable.
- Line 84; the sentence sounds scientifically inappropriate as it reads to me that the virus destroys the cells immediately after the latency is switched to the lytic phase. In my opinion, it should be modified to “during EBV lytic cycle, the cellular machinery is used to replicate and translate viral components including viral proteins in order to produce virions and, in the process, destroys the cell.”
- Line 85; although it is established that EBV virions produced from B cells show tropism towards epithelial cells and vice versa, the virions can still infect both B cells and epithelial cells.
- Line 96; there is no ‘evolution’ in B cells and therefore I suggest either the authors revise the term 'evolutionary' or explain it.
These are just a few examples in the review where I feel need to be revised. There are many more statements in this paper which are scientifically weak and need to be amended with appropriate references, before this gets published in this journal.
Reviewer 2 Report
In this review, the authors introduced the cellular and molecular composition of TME in EBV-infected cells, immune evasion strategies for different EBV-associated malignancies, and the potential immunotherapy by targeting the TME. It is better to further summarize the immunotherapies that are used for treatment of EBV-associated malignancies. Moreover, several comments are listed below for the authors’ consideration.
Please provide the specific references to support their descriptions, including:
Line 46. “Males encounter a higher EBV infection compared to females.”
Line 46-48. “Geographically, EBV-associated malignancy is present all over the world but is more prevalent in Asia and Africa than in the Western world.”
Line 57. “Primary EBV infection often involves children silently before the age of 5 years.”
Figure 1. The authors summarized the EBV life cycles occur in these corresponding B cells, for example, latency III in naïve B cells, latency II in germinal center cells, and so on. This may not be convincing in the EBV field, and please list the supporting evidence if they insist on the statements.
Please double-check and use consistent terminology in this review. For example, EBNA1 is typed as EBNAI by mistake in several places.
Round 2
Reviewer 1 Report
The manuscript is acceptable to me in the current format.
Reviewer 2 Report
All of my comments are properly addressed by the authors, so this manuscript is suitable to be published in the journal.